# Communicative modulations of early action components support the prediction of distal goals

**Martin Dockendorff** [1] *, **Laura Schmitz**[2], **Cordula Vesper**[3,4], **Günther Knoblich**[1]

**1** Department of Cognitive Science, Central European University, Vienna, Austria, **2** Department of Neurology, University Medical Center Hamburg-Eppendorf, Hamburg, Germany, **3** Department of Linguistics, Cognitive Science, and Semiotics, Aarhus University, Aarhus, Denmark, **4** Interacting Minds Centre, Aarhus University, Aarhus, Denmark

* dockendorff_martin@phd.ceu.edu

**Data Availability Statement:** All data files and videos are available on OSF under the following link: https://osf.io/pv74b/.

**Funding:** "This research was supported by the European Research Council under the European

## Abstract

The successful unfolding of many social interactions relies on our capacity to predict other people's action goals, whether these are proximal (i.e., immediate) or distal (i.e., upcoming). The present set of studies asks whether observers can predict the distal goal of two-step action sequences when presented with communicative modulations of the first movement component of the sequence. We conducted three online experiments in which we presented participants with animations of a box moving to a first target location before moving onwards to a final, either near or far, target location. The second movement component and the target locations were occluded. After observing the first movement, participants were asked to select the most likely final target location, i.e., the distal goal of the sequence. Experiment 1 showed that participants relied on the velocity modulations of the first movement to infer the distal goal. The results of Experiment 2 indicated that such predictions of distal goals are possible even when the second movement in the sequence does not contain any velocity information, thus suggesting that the information present in the first movement plays the major role in the process of linking movements to their distal goals. However, Experiment 3 showed that under some circumstances the second movement can also contribute to how observers predict a distal goal. We discuss these results in terms of the underlying simulation processes that enable observers to predict a distal goal from the observation of proximal communicative modulations.

## Introduction

Many social interactions rely on our capacity to predict other people's actions and to quickly adapt our behavior accordingly [1]. Even a simple social act like shaking hands requires one to anticipate and monitor the other person's arm and hand movements so as to make one's palm meet the exact same spot on the other person's hand—all this in a matter of just a few seconds [2]. Such simple joint actions, and also more complex ones such as playing a piano duet, bene-fit from (and sometimes are even made possible by) our capacity to predict the outcome and

Union's Seventh Framework Program (FP7/2007-2013) / ERC grant agreement n° [609819], SOMICS. There was no additional external funding received for this study. The funders had no role in the study design, data collection and analysis, decision to publish, or preparation of the manuscript."

**Competing interests:** The authors have declared that no competing interests exist.

timing of others' actions while these unfold [3], or even prior to their initiation [4]. In order to facilitate such predictions, interacting agents will often resort to a variety of behavioral strategies, aptly known as "coordination smoothers" due to their role in simplifying coordination demands during interaction [5, 6]. One such smoother involves the modulation of certain kinematic parameters of an instrumental action in order to make the action more salient and readable to an observer, who can then predict the action outcome more easily. For example, one of the pianists in the duo might lift her arms with a high amplitude right before starting to play, thus providing more explicit anticipatory information about the timing of her immediate actions to her co-performer [7]. Crucially, such modulations are characterized by the fact that their underlying goals have a "dual nature" [8]: the person performing the action can achieve simultaneously an instrumental goal (e.g., pressing one of the keys on the piano) and a communicative goal (e.g., informing the co-performer about the exact moment when she will start playing) [9]. In keeping with previous research showing how such communicative modulations can be used to facilitate coordination [10], we will refer to these as "sensorimotor communication" (henceforth 'SMC').

Most previous research on SMC has focused on how actors coordinate their actions by relying on communicative modulations of instrumental actions directed to proximal goals [see 8 for a review]. Proximal goals can be understood as those goals whose achievement is the result of a single transitive or intransitive movement, like placing an object on a table or making a step to a new position in space, respectively. Proximal goals are therefore directly tied, both temporally and spatially, to the movements leading to their immediate achievement. Distal goals, on the other hand, are achieved by two or more successive movements, each with its own proximal sub-goal. Consequently, and unlike proximal goals, the early movement(s) preceding the achievement of a distal goal are both spatially and temporally separated from it by one or more intermediate movements. A well-studied example of such distal goals, and one that will be the focus of the present work, are the goals that result from a two-step action sequence, as when a grasping movement towards a piece of food (i.e. a first movement component) is followed by bringing it to one's mouth (i.e., a second movement component) [11, 12].

The present set of studies draws on two lines of research: first, research on people's production and understanding of actions directed at distal goals, and second, research on SMC. Based on these two lines of research, we ask under which conditions observers can predict the distal goal of a two-step action sequence when observing communicative modulations of a first movement component. Before addressing this question, we will first review relevant findings in the motor control literature suggesting that acting towards distal goals leads to changes in kinematic features of early movement components [13]. These changes in movement kinematic can be used by observers to predict distal goals of observed actions [14]. Then, we will argue that such predictive processes play a key role in SMC, where observers use communicative modulations of instrumental actions to simulate distal goals, thereby establishing what we call "motor-iconic" relations between modulations in the kinematics of early action components and the action's distal goal.

## Distal goals in action production and observation

A large body of research in motor control has demonstrated that the way we plan and execute simple motor acts in non-communicative contexts is highly sensitive to our upcoming actions and to the distal goals achieved by these actions [15]. For example, a natural reaching movement towards an object is performed with different velocities depending on whether the reach is followed by a careful placing of the object or by throwing the object in a large box [12, 16]. Similar effects of distal goals on early action kinematics have been found when participants are

asked to reach towards an object and then place it in a target location that varies in size, position or relative distance [13]. Specifically, in the latter study, reaching velocity was higher and maximal finger aperture was larger when the object had to be placed in a far target relative to a near one [see also 17, 18]. Furthermore, studies on the "end-state comfort effect" report a tendency for participants to grasp objects in bio-mechanically awkward ways in order to ensure a comfortable position at the end of their movements [19, 20], thus again suggesting a strong influence of distal goals on the planning and execution of early movement components.

One way researchers have sought to explain how distal goals affect early action kinematics is by appealing to the mediating role of motor or action representations [21–23]. These representations play a key role in guiding actions towards their goals while the action unfolds [11, 24]. As a consequence, performing an action directed at a distal goal, like throwing an object into a basket after picking it up, can sometimes lead to visible changes in the kinematics of the early movement components (in this case, the grasping of the object) leading towards such goal (for a similar proposal in terms of "coupled planning" see [25]). Other features of distal goals, such as their expected value (i.e., reward), can also lead to visible changes in early movements and, in turn, increase people's motor performance [26, 27].

Besides their role in guiding the performance of one's own actions, motor or action representations are also involved in understanding and predicting other people's instrumental actions and goals [28–30]. Crucially, these predictions often go beyond the mere online anticipation of an unfolding movement or its proximal (i.e., immediate) goal, as they can extend to the prediction of more distal goals [14]. For example, research suggests that motor regions in the brain are differently activated depending on whether an observed reaching movement towards a piece of food is then followed by the distal goal of placing it in a container or of eating it [31]. What makes these predictions possible is the fact that observers rely on early kinematic information contained in the movements [14, 32–34] but also on objects placed in the immediate vicinity of the agent [35, 36].

Most research on action understanding has shown that observers can use kinematic information as a cue to infer information about the observed actor's (both proximal and distal) goals. This has recently led researchers in the field of joint action to ask whether, in social interactions, co-actors would not only rely on each other's kinematic information to coordinate their actions, but would also actively exaggerate their movements, thus facilitating the achievement of a joint goal by making their action goals easier to predict [32, 33, 37–41]. These situations, in which co-actors actively inform each other about their goals by modulating the kinematics of their movements, are at the core of SMC. We turn to these now.

## Proximal and distal goals in SMC

Studies on SMC have mostly focused on settings in which co-actors are required to coordinate their actions by, for example, aiming at one of three target locations either in synchrony [42] or sequentially [43]. Typically, the design of these studies is such that the information allowing co-actors to achieve coordination, like the location of the correct target, is allocated to one of the actors only, the "Leader". Furthermore, participants are not allowed to speak nor gesture during the performance of these tasks. Instead, Leaders inform their naïve "Followers" by modulating key kinematic features of their instrumental actions. For example, Leaders have been shown to exaggerate the amplitude of their aiming movements in order to disambiguate between target locations [42], or to increase or decrease their wrist height as a way of informing the Follower about the part of the object they are about to grasp [44]. These modulations are more easily perceived by Followers, who can use them to disambiguate between various proximal goals [42, 45–47]. Once they are able to detect and disambiguate between different

proximal goals, Followers can adapt their behavior in a timely manner, thereby leading to successful temporal and/or spatial coordination [43, 48]. Thus, given that Leaders exaggerate their movements (in studies on SMC), and that these exaggerations are in turn used by Followers to facilitate their predictions of the Leader's goals, these movements have sometimes been described as "signals" [8], to be contrasted with the natural, non-exaggerated movements in studies on action observation, where observers rely on these movements as "cues" to predict the goals of the agent.

At the core of SMC is the capacity for Leaders to make their proximal goals more discriminable by producing kinematic modulations, and for Followers to understand such modulations as conveying anticipatory information about these goals. However, as we noted previously, the predictions made during action observation are not limited to proximal goals, but are also directed at more distal goals. Thus, this raises the question of whether communicative modulations of early action components can be used by observers to predict an agent's distal goals.

Some of the findings on action observation we reviewed earlier provide preliminary evidence indicating that observers can derive information about distal goals when observing the initial stages of naturally performed reaching movements [49, 50]. Thus, these studies indicate that observers can infer distal goals when observing instrumental movements that are not communicatively modulated by their actors. As far as we know, only two studies have directly tested whether observers can also do this for communicative modulations of actions. Donnarumma and colleagues [51] conducted a series of computationally-guided analyses on the processes involved in the recognition of two-step action sequences. Their analyses suggest that a performing agent who actively modulates her first movement component so as to increase the similarity in its kinematic features with respect to a second movement component [by means of "coarticulation", see 52] can facilitate an observer's early recognition of a distal goal.

In our own previous study [53] we asked directly whether observers would be able to understand kinematic modulations of a first movement in terms of distal goals. To do so, we presented participants with animations of a box sliding at different velocities towards an intermediate target location, after which the box was automatically delivered towards one of two final, occluded target locations, far or near. Participants were then asked to select the final location towards which they thought the box had been delivered. The findings indicated that participants benefit from modulations in velocity of the sliding movement to infer the distal goal. This was indicated by their capacity to consistently map the different movements onto different target locations. Specifically, when participants observed movements with higher peak velocity they were more likely to map them onto the "far" target location, whereas movements with lower peak velocity were more likely to be mapped onto the "near" target location.

We referred to this particular relation holding between modulations in movement velocity and distal goals as "motor-iconic", as it corresponds to a regular relationship between the velocity and distance of unconstrained aiming or grasping movements towards targets of varying distance in either one-step [15] or two-step action sequences [13]. In these studies, participants tend to aim with higher velocity at far compared to near targets. As a consequence, we apply the notion of "motor-iconicity", which refers to the regular relationship between non-communicative movements and goals as found during natural performance, to the relationship between exaggerated movements and their goals in the context of SMC.

## Two mechanisms for simulating a distal goal

Given that motor-iconic relations capture a regular relationship between communicative modulations of actions and their distal goals, it is unclear how such relationship is established when

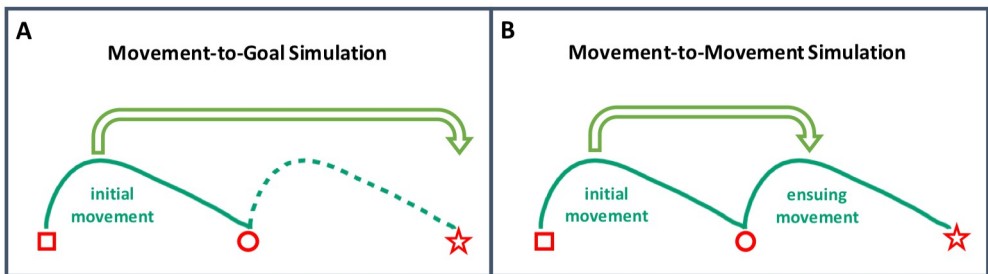

**Fig 1. Two kinds of simulation enabling participants to link communicative modulations of an initial movement to a distal goal.** The green lines are a schematic representation of velocity profiles of sliding movements. The red square indicates the starting point of the initial movement, the red circle both the endpoint of the initial movement and the starting point of the ensuing movement, while the red star indicates the endpoint of the ensuing movement, i.e., the distal goal of the sequence. **A**. In the movement-to-goal simulation, observing an initial movement is sufficient to predict the distal goal of the action, and can therefore bypass the simulation of the ensuing movement. **B**. In the movement-to-movement simulation, the initial movement enables an observer to simulate an ensuing movement, which in turn enables the simulation of the distal goal.

observing two-step action sequences. Based on the results of Dockendorff et al. [53], we can conclude that such relationship can be captured by establishing a "direct" link between a first movement component and a distal goal. That is, a first movement component containing sufficient kinematic information can be used to predict an upcoming distal goal without relying on a second movement component. This implies that it is sufficient to observe modulations in the initial movement component to predict distal goals. Since such a process relies on linking kinematic information from an initial movement directly to its distal goal, we will refer to it as a "movement-to-goal" simulation (see Fig 1A).

Another possibility is that a second movement component within a sequence plays an active role in the process of linking communicative modulations of first movement components to distal goals. In such scenarios, the initial movement is fed into a simulation of an ensuing movement, rather than into a direct simulation of the distal goal. Since this process involves taking into account the mediating role of the second movement component in the sequence, we will refer to it as "movement-to-movement" simulation (see Fig 1B).

Note that we use the term "simulation" here to refer to the capacity observers have to link movements to distal goals. This process of linking movements and goals might rely on (online) motor mirroring, as proposed by research on action understanding and SMC [4, 10]. However, in the context of our experiment, it is also possible that participants built these relations based on other processes, such as action-effect associations [54], generalized mechanisms of statistical learning [55] or even on Gestalt-like principles applied to action recognition [56].

Besides the two types of simulation discussed above, it is also possible that observers do not engage in any type of simulation but that, once they identify that the movements contain some relevant kinematic information ("communicative signals"), they try to establish a stable link between them and the distal goals. This mapping is arbitrary in the sense that the observer makes a choice at the beginning of the experiment and does not engage in a simulation (and so does not necessarily choose a mapping that would be seen as motor-iconic). This form of arbitrary interpretation contrasts with a situation in which an observer disregards the communicative signals present in the movements altogether, in which case one would expect them to link the movements to their distal goals in a random fashion.

To address these different possibilities, we present three experiments in which participants needed to infer a distal goal on the basis of modulations in velocity and/or duration of a first

movement within a two-step sequence. The purpose of Experiment 1 was to investigate how different kinematic information present in the first movement component of the sequence would be then fed into a simulation of a distal goal. Then, in Experiments 2 and 3 we address the question of whether participants interpret communicative modulations of a first movement component in a way that takes into account the second movement component. Thus, we were interested in determining whether participants rely on movement-to-goal simulations, or on movement-to-movement simulations (or on both).

## Experiment 1. What kinematic information is needed for a simulation of a distal goal?

The aim of the first experiment was to identify how different kinematic information present in a first movement component helps participants to simulate its distal goal. To do so, we compared two-step action sequences in which the first movement component contained continuous velocity information (Sliding-Sliding condition) or only discrete duration information (Jumping-Sliding condition). The initial movement was then followed by a continuous movement towards an occluded, and therefore unknown, target location. If the presence of velocity and/or duration information facilitates the prediction of a distal goal, participants should consistently map fast initial movements onto far targets and slow initial movements onto near targets, consistent with previous findings in motor control [13]. Moreover, we expect participants to benefit from the exaggerations in velocity and/or duration, as such exaggerations generally facilitate the prediction of goals during SMC [10].

### Methods

**Participants.** We recruited 50 participants (29 women; Age: M = 30.8 years; SD = 9.6 years), 25 per condition, through the online testing platform Testable (https://www.testable.org/). All participants gave their informed written consent prior to inclusion in the study, in accordance with the Psychological Research Ethics Board (PREBO; reference number 2020_04). Sample size was determined using the Superpower statistical package [57] on RStudio [58]. The design and analyses of the study were preregistered on OSF (https://osf.io/px96b). The videos used in all experiments are publicly accessible on OSF, at https://osf.io/pv74b/. Data collection for all studies was performed between May 2021 and November 2022.

**Stimuli.** The basic layout for both experimental conditions of Experiment 1 is shown in Fig 2. In both conditions, participants were first presented with a stationary box with a mouse cursor attached to it. The box and cursor were displayed within a black hexagonal location on the left-hand side of the screen. During familiarization, participants also saw a grey hexagonal intermediate location in the middle of the screen and two green hexagonal target locations on the right-hand side of the screen. A black horizontal line, along which the box moved during the trials, connected the initial location to the grey hexagonal area in the middle of the screen and to the two green target locations in both conditions (see Fig 2).

During trials, a black occluder covered different sections of the display. During the first movement component (i.e., when the box moved from the initial location to the grey intermediate target location), the occluder only covered the green target locations (Fig 2A). Right before the beginning of the second movement component (i.e., before the box started sliding from the intermediate target location towards the green target locations) the occluder was widened, thus covering the intermediate grey target and the box itself (Fig 2B). This prevented participants from seeing the initial acceleration phase of the second sliding movement, from which they could have derived information about where the box would then move (i.e., the near or far target).

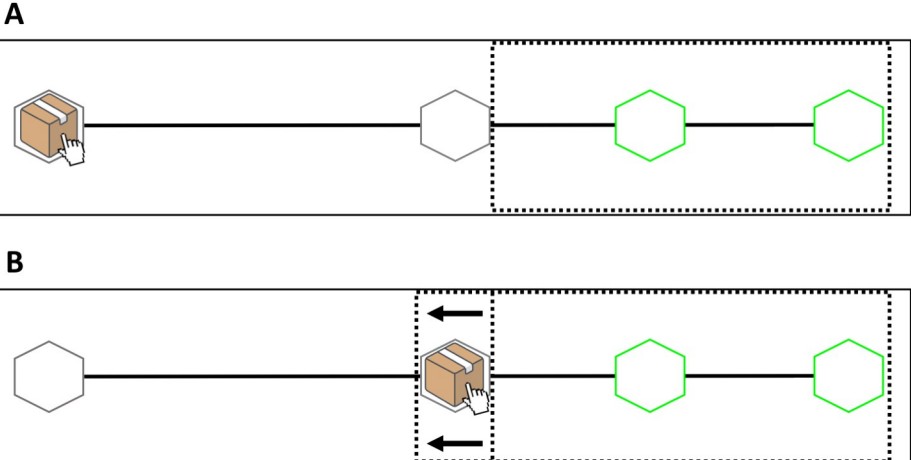

**Fig 2. Experimental layout used in Experiment 1.** The black dotted line represents the outline of the occluded area in both conditions during trials. **(A)** At the beginning of the trial, the occluder only covers the green target locations, leaving visible the grey target location in the middle of the screen. **(B)** Once the box had arrived at the intermediate target location, and right before it started sliding towards the green targets, the occluder became wider in order to cover the beginning of the second sliding movement.

**Design.** The experiment consisted of a mixed 2x3 factorial design, with one between-subject factor (first movement type) and one within-subject factor (degrees of exaggeration). The between subject factor manipulated whether the movement of the box from the initial to the intermediate location was presented as a continuous, fully visible sliding movement between these two locations (i.e., Sliding-Sliding condition) or whether the same movement was presented as an invisible, discrete "jumping" movement from the origin to the intermediate location with a duration that matched the total duration of the corresponding sliding movements (i.e., Jumping-Sliding condition). The within-subject factor manipulated whether and to what degree the animated movements were exaggerated in terms of peak velocity or duration (i.e., Normal (i.e., no exaggeration), Exaggerated, Very exaggerated). The procedure used to create such movements is the same used in Dockendorff et al., [53]. Briefly, it consisted in exaggerating two non-exaggerated movements by adding or subtracting two predefined values from their peak velocities, thus yielding a total of six movements: the two original non-exaggerated ones (i.e., Normal), a slow and a fast one (i.e., Exaggerated), and a very slow and a very fast one (i.e., Very Exaggerated). Since participants in the Jumping-Sliding condition only saw a snapshot of the movement before onset and after offset, these exaggerations were never perceived as such. Instead, participants could only rely on the differences in total duration, corresponding to the time interval between the disappearance and reappearance of the box.

## Procedure

**Familiarization and instructions.** During the familiarization, participants were first presented with the complete task layout (as illustrated in Fig 2), but without the occluder covering the green target locations. Participants then saw two successive Normal movements of the box towards the middle target, where it stayed stationary for approximately 1500 ms, after which the box started sliding towards either of the two target locations (order counterbalanced across participants). After seeing the two Normal movements, participants in both conditions were asked to select the target location where the box had moved by pressing the "n" key (for near) or "f" key (for far). In the Jumping-Sliding condition, participants were also told at the

very beginning of the experiment that they would "only see a snapshot of the box disappearing at the starting position and then a snapshot of it reappearing in the middle grey target. Thus, you will not see the actual sliding movement performed by the previous participant, but only the beginning and end of it."

Next, a black occluder covered the target locations (Fig 2A) and participants were presented with two more Normal movements of the box towards the middle target (order counterbalanced). Right before sliding from the middle target to the (now occluded) target locations, the occluder became wider on the left side, thus covering the box. Participants were explicitly told that this change in occluder size indicated the onset of the second sliding movement (Fig 2B). Finally, participants were told that the movements they would see had been recorded from a previous participant, and that this participant had produced such movements "in ways that would help others guess to which green target location he/she was moving the box".

**Experiment.**   Participants performed 36 experimental trials, divided into six blocks. In each trial, they were presented with an animation of the box either sliding along the black line in the Sliding-Sliding condition or disappearing from the origin and reappearing at the intermediate location in the Jumping-Sliding condition. Participants in both conditions were then prompted to answer to which target location they thought the box had been delivered. Note that in both conditions the second sliding movement was never visible to participants during the trials, but, as explicitly stated in the familiarization, its onset was indicated by the widening of the occluder. A trial was completed when participants pressed one of the two assigned keys ("n" or "f"), corresponding to either the "near" or "far" target locations. Each block contained all six degrees of exaggeration, presented in random order. Participants did not receive feedback about their performance at any point. At the end of the experiment, participants were asked to fill out a short questionnaire about their experience with the task.

## Results

**Data preparation.**   We categorized participants' responses as *Iconic* or *Non-Iconic* mappings. Iconic mappings refer to trials where participants pressed the "n" key in response to movements with lower peak velocity (and thus longer total duration) and the "f" key in response to movements with higher peak velocity (and thus shorter total duration). Non-Iconic mappings, on the other hand, refer to those trials where participants reversed this association, i.e., by pressing "f" in response to movements with lower peak velocity (i.e., longer total duration) and "n" in response to movements with higher peak velocity (i.e., shorter total duration).

Two dependent variables were computed from participants' number of Iconic and Non-Iconic mappings. Calculating the *absolute* difference between the total number of Iconic mappings and the total number of Non-Iconic mappings, separately for each first movement type and each degree of exaggeration, yielded a *Consistency score* for each participant ranging from 0 to 12. A score of 0 meant that participants mapped velocities and/or durations randomly to targets and a score of 12 meant that participants mapped with absolute consistency. Calculating the *signed* difference between Iconic and Non-Iconic mappings gave us the *Mapping score*, which could range from +12 (fully iconic mappings) to -12 (fully non-iconic mappings). A Mapping score of 0 meant that participants had no preference for any of the two mapping directions.

Participants who pressed the same key (either "n" or "f") at least ten times in a row were excluded from further analysis. Based on this criterion, 4 participants were excluded in Experiment 1.

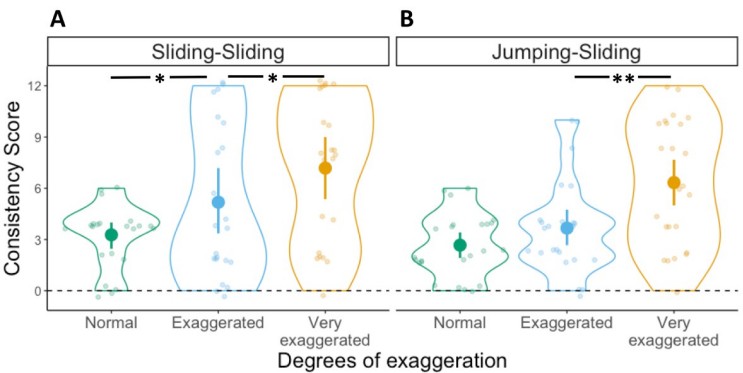

**Fig 3. Consistency scores in Experiment 1.** Distribution of Consistency scores in the (A) Sliding-Sliding and (B) Jumping-Sliding conditions, for the three degrees of exaggeration. Each dot represents an individual participant, with one Consistency score for each degree of exaggeration: Normal in green, Exaggerated in light blue and Very exaggerated in yellow. Violin plots represent the overall distribution of Consistency scores for each degree of exaggeration. The dashed horizontal line indicates a hypothetical value for random mapping (i.e., no consistency). Significant differences between degrees of exaggeration and across conditions as derived from a post-hoc t-test are indicated by horizontal lines and asterisks above the violin plots (* = $p \leq 0.05$; ** = $p \leq 0.01$; *** = $p \leq 0.001$).

**Consistency score.** To test whether participants benefited from the exaggerations by producing more consistent mappings, and whether such effect differed depending on whether velocity information was present in the movements, we conducted a 2x3 ANOVA with Consistency scores as dependent variable, first movement type (Sliding, Jumping) as between-subject factor and degrees of exaggeration (Normal, Exaggerated, Very exaggerated) as within-subject factor. The ANOVA yielded a main effect of degrees of exaggeration ($F(2,88) = 23.4$, $p < .001$, $\eta^2 = .19$), but no main effect of first movement type ($F(1,44) = 1.8$, $p = .18$, $\eta^2 = .023$). The interaction between these two factors was not significant ($F(2,88) = .36$, $p = .7$, $\eta^2 = .003$) (see Fig 3). This pattern of results suggests that regardless of whether the first movement contained velocity information (i.e., Sliding) or not (i.e., Jumping), participants were able to consistently map the different movements to the target locations ($M$ Consistency Score: Sliding-Sliding = 5.21, Jumping-Sliding = 4.22) and benefited from their exaggerations.

**Mapping scores.** A 2x3 ANOVA was conducted with Mapping scores as dependent variable, first movement type as between-subject factor and degrees of exaggeration as within-subject factor. The ANOVA revealed a main effect of first movement type ($F(1,44) = 20.1$, $p < .001$, $\eta^2 = .22$), but no main effect of degrees of exaggeration ($F(2,88) = 2.3$, $p = .11$, $\eta^2 = .02$). Furthermore, we found a significant interaction between these two factors ($F(2,88) = 7.6$, $p < .001$, $\eta^2 = .06$). To further explore the main effect of first movement type we conducted Bonferroni-corrected $t$-tests comparing the Mapping score distributions for each degree of exaggeration across the two between-subject conditions (e.g., Exaggerated in Sliding-Sliding versus Exaggerated in Jumping-Sliding). The comparison of Mapping scores across conditions between Normal movements did not reach significance ($t(94) = 1.3$ $p = .18$, $d = 0.6$), but the one between Exaggerated ($t(94) = 4$, $p < .001$, $d = 1.2$) and Very Exaggerated movements ($t(94) = 5.3$, $p < 001$, $d = 1.2$) did (see Fig 4). This suggests that participants benefit from communicative modulations of peak velocity (i.e., Sliding-Sliding) since they are able to map the movements to the targets in a motor-iconic fashion. Modulations in movement duration (i.e., Jumping-Sliding), however, did not lead to more motor-iconic mappings.

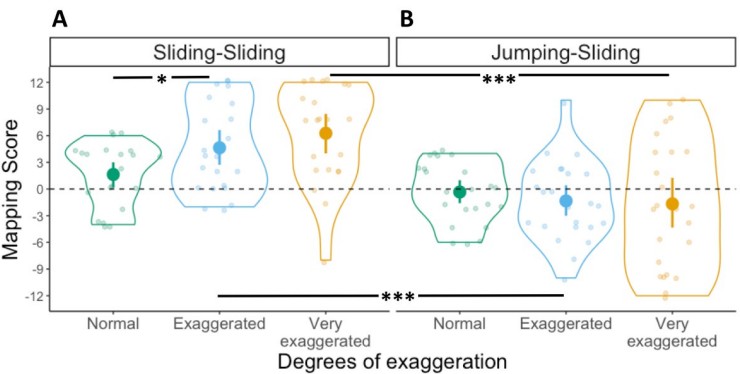

**Fig 4. Mapping scores in Experiment 1.** Distribution of Mapping scores in the (A) Sliding-Sliding and (B) Jumping-Sliding conditions, for the three degrees of exaggeration. Each dot represents an individual participant, with one Mapping score for each degree of exaggeration. Violin plots represent the overall distribution of Mapping scores for each degree of exaggeration. The dashed horizontal line indicates random mapping direction. Significant differences between degrees of exaggeration and across conditions as derived from a post-hoc t-test are indicated by horizontal lines and asterisks above the violin plots (* = $p \leq 0.05$; ** = $p \leq 0.01$; *** = $p \leq 0.001$).

## Discussion

The general aim of Experiment 1 was to identify how different kinematic information present in a first movement component within a two-step action sequence is used by observers to infer its distal goal. We were particularly interested in determining whether the presence of continuous velocity information or of discrete duration information presented in the communicative modulation of the first movement of the sequence would enable participants to simulate the distal goal.

Participants in the Sliding-Sliding condition, who saw the box sliding with different peak velocities, were able to use the differences in velocity to simulate the distal goal of the action sequence, as indicated by the high consistency of their mappings (Fig 3) and the fact that such mappings were motor-iconic, that is, in line with the underlying regular relation between movements and goals found in natural movement performance (Fig 4) [15, 13]. This contrasted with the Jumping-Sliding condition, in which participants were also highly consistent in their responses, but less likely to produce motor-iconic mappings. This indicates that in both conditions, participants were able to create stable mappings based on the communicative modulations present in the movements, but that they relied on motor-iconic mappings only if the first movement contained continuous velocity information. When the initial movements only contained discrete information about their total duration, as in the Jumping-Sliding condition, this was less likely to trigger the appropriate simulation process that enables the prediction of the distal goal.

One possible explanation for the lower motor-iconic mappings in the Jumping-Sliding condition is that the lack of velocity information introduces, among some participants, a more arbitrary interpretation of the relationship between movements and distal goals. Since this arbitrary relation is the result of participants simply choosing a particular link between movement duration and target location, some of them chose a link that reversed the motor-iconic mapping, thus mapping faster peak velocities to the near target and lower peak velocities to the far target. Our results of the Jumping-Sliding condition provide support for this hypothesis, given that participants are able to map the movements onto the target locations consistently (see Fig 3B), but they fail, collectively, to display a unique mapping preference (see Fig 4B),

since they were sometimes nearly split between motor-iconic mappings and their reversal (Motor-iconic: N = 10; Reversed: N = 13; see Very Exaggerated in Fig 4B).

Interestingly, what may have led participants to reverse the mapping in the Jumping-Sliding condition is an altogether different intuition about the relationship between movement duration and target distance. The intuition is nicely captured in the following description given by one of our participants in the Jumping-Sliding condition who, when asked to explain her strategy to solve the task, replied: "just assuming that [the] movement that *took longer* was supposed to indicate a *longer journey*" (italics ours). In other words, according to this intuition, longer movements (i.e., movements with longer total durations, and thus, in our task, with lower peak velocities) are more likely to travel longer distances than shorter movements (i.e., movements with shorter total durations, and thus, in our task, with higher peak velocities). As a consequence, this intuition leads to a reversal of the above specified motor-iconic mapping. This reversal has, interestingly, already been reported in previous studies on SMC in which participants used total duration to communicate information about target distance [43]. We return to a more detailed discussion of this mapping reversal in the General Discussion.

One final explanation for the lower motor-iconic mappings in the Jumping-Sliding condition would be that participants simply fail to perceptually discriminate the differences in duration between the movements. However, the Consistency scores in that condition show that participants are able to create stable associations throughout the experiment between movement durations and target locations, regardless of how exaggerated they are. Participants could not form such stable associations if they were not able to discriminate the differences in movement duration [53].

In sum, our findings indicate that observers rely on velocity information to infer and simulate a distal goal. However, this pattern is consistent with both types of simulation we propose (see Fig 1). Concretely, participants in the Sliding-Sliding condition may have either used the velocity of the initial movement to simulate the velocity of the upcoming one, and subsequently the distal goal (i.e., movement-to-movement simulation), or they may have disregarded the upcoming movement and simulated the distal goal directly based on the first movement only (i.e., movement-to-goal simulation). In order to gain a clearer understanding of which type of simulation is being used, we conducted Experiment 2.

## Experiment 2. What kind of simulation underlies the prediction of a distal goal?

In Experiment 2, participants saw a first sliding or jumping movement to the intermediate location (as in Experiment 1), but this time were familiarized with a second jumping movement rather than a second sliding movement as in Experiment 1. This meant that the first movement (Sliding or Jumping) was now followed by a second movement that, unlike Experiment 1, did not contain velocity information, but only duration. Accordingly, we labeled the two conditions of the present experiment Sliding-Jumping and Jumping-Jumping.

If participants are engaging in movement-to-goal simulations, then we expect them to be able to simulate the distal goal only when the observed movement contains continuous velocity information, and regardless of whether the second movement contains velocity information or not. Thus, we expect participants in the Sliding-Jumping condition to map more consistently in the motor-iconic direction than in the Jumping-Jumping condition, and thereby produce a pattern of responses similar to the one found in the Sliding-Sliding condition of Experiment 1. In contrast, if participants engage in movement-to-movement simulations, which take into account the presence (even if implied) of a second continuous movement in the process of simulating the distal goal, we expect participants in the Sliding-Jumping condition to have

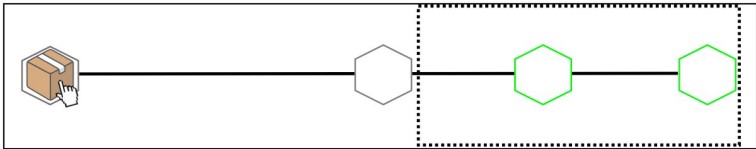

**Fig 5. Experimental layout used in Experiment 2.** Experimental layout in the Sliding-Jumping and the Jumping-Jumping condition. The black dotted line represents the outline of the occluded area during trials. The (near and far) target locations are displayed in light green and the intermediate location is displayed in grey.

difficulties in simulating the distal goal, and therefore produce less motor-iconic mappings than in the Sliding-Sliding condition of Experiment 1. More generally, we expected participants to be able to use the communicative modulations of movement velocity and/or duration, and therefore to benefit from their exaggerations by producing more consistent mappings.

## Methods

**Participants.** We recruited 50 participants (25 women; Age: $M$ = 29.8 years; $SD$ = 9.9 years) through Testable. The conditions of recruitment were identical to Experiment 1. Based on the same exclusion criteria used in Experiment 1, we excluded 2 participants from our analyses. The design and analyses of this study were pre-registered on OSF (https://osf.io/upg92).

**Stimuli.** The layouts for each experimental condition of Experiment 2 were very similar to Experiment 1 (see Fig 5). The only difference with respect to Experiment 1 pertained to the size of the occluder, which in both conditions was kept the same size throughout the trial, thus leaving the intermediate target always visible.

**Design and procedure.** As in Experiment 1, Experiment 2 consisted of a 2x3 mixed-factorial design with first movement type as a between-subject factor and degrees of exaggeration as a within-subject factor.

Participants were familiarized with the full layout and were presented with two normal movements. Once the box reached the intermediate location, where it stayed stationary for approximately 1.5 secs, it then disappeared and reappeared in one of the two target locations (order counterbalanced across participants). Note that the duration of these second jumping movements matched the duration of the second sliding movements used in the familiarization of Experiment 1. Thus, the duration of the second jumping movement directed to the far target was 2.74 secs, while the one to the near lasted 2.23 secs. After seeing the two Normal movements, participants in both conditions were asked to select the target location where the box had moved by pressing the "n" key (for near) or "f" key (for far). Then, they were presented with two more Normal movements, but this time the green target locations were covered with the black occluder (Fig 5).

**Experiment.** In each trial, participants were presented with an animation of the box either sliding along the black line in the Sliding-Jumping condition or disappearing from the origin and reappearing at the intermediate location in the Jumping-Jumping condition. Unlike Experiment 1, however, participants saw the box disappearing from the middle gray target, which indicated the beginning of the second jumping movement. Participants then selected the target location to which they thought the box was delivered by either pressing the "n" key (for near) or "f" key (for far).

## Results

**Consistency score.** We conducted a 2x3 ANOVA with Consistency scores as dependent variable, first movement type as between-subject factor and degrees of exaggeration as within-

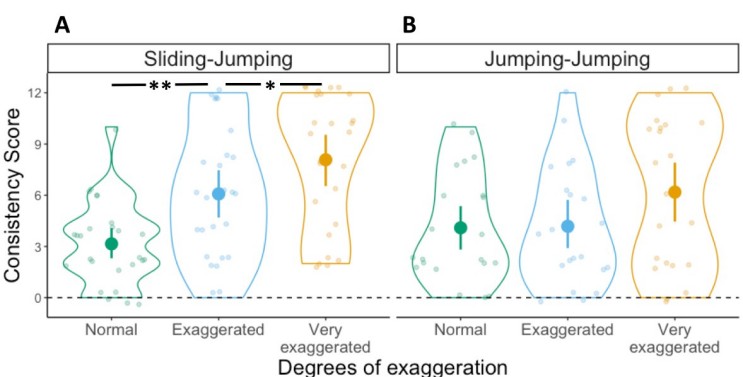

**Fig 6. Consistency scores in Experiment 2.** Distribution of Consistency scores in the (A) Sliding-Jumping and (B) Jumping-Jumping conditions. Violin plots represent the overall distribution of Consistency scores for each degree of exaggeration. The dashed horizontal line indicates a hypothetical value for random mapping (i.e., no consistency). Significant differences between degrees of exaggeration and across conditions as derived from a post-hoc t-test are indicated by horizontal lines and asterisks above the violin plots (* = $p \leq 0.05$; ** = $p \leq 0.01$; *** = $p \leq 0.001$).

subject factor. The ANOVA yielded a main effect of degrees of exaggeration ($F(2,92)$ = 16.8, $p$ < .001, $\eta^2$ = .14), and an interaction between degrees of exaggeration and first movement type ($F(2,92)$ = 3.6, $p$ = .03, $\eta^2$ = .035). However, there was no significant main effect of first movement type ($F(1,46)$ = 1.5, $p$ = .22, $\eta^2$ = .03). This pattern of results suggests that participants benefited from communicative modulations of movements containing continuous velocity information (i.e., Sliding-Jumping, $M$ = 5.77), but not necessarily from modulations of movements containing only discrete duration information (i.e., Jumping-Jumping, $M$ = 4.82) (see Fig 6).

**Mapping score.** We conducted a 2x3 ANOVA with first movement type as a between-subject factor and degrees of exaggeration as a within-subject factor. The ANOVA yielded a main effect of first movement type ($F(1,46)$ = 9.6, $p$ = .003, $\eta^2$ = .12) but not of degrees of exaggeration ($F(2,92)$ = 1.8, $p$ = .16, $\eta^2$ = .01). The interaction between these two factors did not reach significance ($F(2,92)$ = 2.1, $p$ = .13, $\eta^2$ = .01). This replicates our findings of Experiment 1 and confirms that participants are more likely to map movements motor-iconically if the first movement contains continuous velocity information rather than discrete duration one (see Fig 7).

**Type of simulation.** To see whether the presence of a second movement that contains velocity information plays a role in simulating a distal goal, we conducted a between-Experiment comparison. Specifically, we conducted a 2x3 ANOVA using the Mapping scores of the two conditions in which participants saw velocity information in the first movement component (i.e. a Sliding movement: *Sliding*-Sliding in Exp. 1 and *Sliding*-Jumping in Exp. 2) as a dependent variable. Thus, the between-subject factor in the ANOVA was the second movement type, which in this case either contained velocity information (Sliding-*Sliding* in Exp. 1) or not (Sliding-*Jumping* in Exp. 2). As in our previous analyses, the within-subject factor was the degrees of exaggeration. The ANOVA yielded a main effect of degrees of exaggeration ($F(2,92)$ = 12.5, $p$ < .001, $\eta^2$ = .08), but no main effect of second movement type ($F(1,46)$ = 1.44, $p$ = .23, $\eta^2$ = .02) and no significant interaction ($F(2,92)$ = .27, $p$ = .76, $\eta^2$ = .002).

## Discussion

In Experiment 1 we found that participants were able to rely on communicative modulations of continuous velocity information to simulate the distal goal of the action sequence. When

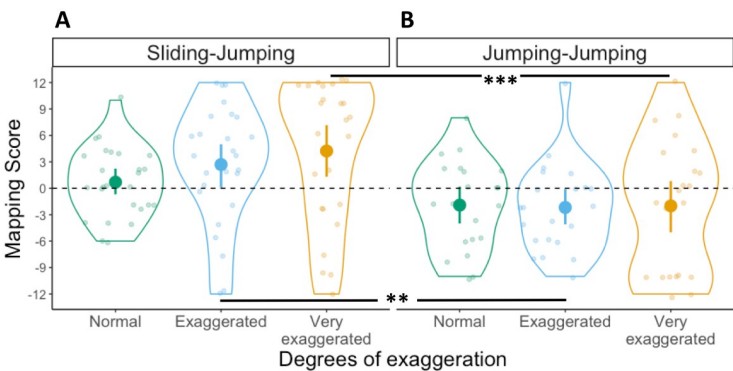

**Fig 7. Mapping scores in Experiment 2.** Distribution of Mapping scores in the (A) Sliding-Jumping and (B) Jumping-Jumping conditions, for the three degrees of exaggeration. Violin plots represent the overall distribution of Mapping scores for each degree of exaggeration. The dashed horizontal line indicates random mapping direction. Significant differences between degrees of exaggeration and across conditions as derived from a post-hoc t-test are indicated by horizontal lines and asterisks above the violin plots (* = $p \leq 0.05$; ** = $p \leq 0.01$; *** = $p \leq 0.001$).

participants only received discrete information about the duration of the movement, they failed collectively to display any strong mapping preference, thus suggesting that at least some participants did not simulate a distal goal with such minimal duration information (or at least not in the way we predicted, i.e., using motor-iconic mappings). These findings, however, do not allow us to draw conclusions about the type of simulation that observers were using to predict the distal goal on the basis of velocity information. To gain more knowledge about these underlying simulation processes, we modified the second movement component in both conditions of Experiment 2, from a continuous movement to a discrete one that did not contain velocity information. This created a situation in which movement-to-movement simulations of a distal goal on the basis of observing an initial movement containing velocity information were made more difficult.

The results of Experiment 2 show that, despite the fact that participants' simulations in the Sliding-Jumping were made more difficult (because there was no continuous second movement to map onto), they were still able to predict the distal goal from observing modulations in movement velocity in that condition. This was indicated by the higher number of motor-iconic mappings and the increasing consistency (as a function of degrees of exaggeration) in the Sliding-Jumping condition compared to the Jumping-Jumping condition. In this latter condition, unexpectedly, participants did not benefit from the exaggeration of movement duration. These results therefore provide a first indication that observers can predict distal goals directly by engaging in movement-to-goal simulations.

A second indication that observers engage in movement-to-goal simulations, and thus might not need to rely on an implied second movement in order to predict a distal goal, is the fact that participants in the Sliding-Jumping condition collectively agreed as much on the motor-iconic mappings as those in the Sliding-Sliding condition of Experiment 1. Thus, this suggests that it is sufficient for observers to observe velocity modulations to simulate its distal goal, and that a second movement does neither interfere nor contribute to such capacity.

However, when looked at more descriptively, our results also point to differences in how some participants interpret the movements when they rely on direct movement-to-goal simulations. Indeed, the mapping scores in the Sliding-Jumping condition of Experiment 2 show that while the majority of participants mapped the movements onto the target locations motor-iconically (N = 19), a small number reversed this mapping (N = 6), thus suggesting that

those participants either arbitrarily chose a mapping at the beginning of the experiment or, as we suggested in our Discussion of Experiment 1, had opposing intuitions about the mapping.

Finally, a comparison between Sliding-Jumping in Experiment 2 with the Sliding-Sliding condition of Experiment 1 showed no significant difference in performance. This leaves open the possibility that the velocity information in the second movement might not always have an effect on participant's capacity to simulate a distal goal (see Figs 4 and 7). In other words, this result is consistent with both direct movement-to-goal simulations and movement-to-movement simulations. In sum, the findings of Experiment 1 and 2 suggest that observers rely on movement-to-goal simulations to link modulations in movement velocity to a distal goal, but that movement-to-movement simulations could in some circumstances also play a role. To test this possibility more directly, and thus to see if there are circumstances in which observers rely on the second movement to simulate a distal goal, we conducted Experiment 3.

## Experiment 3. When do movement-to-movement simulations enable the prediction of a distal goal?

To explore more directly under which circumstances movement-to-movement simulations facilitate motor-iconic interpretations of communicative modulations, we conducted Experiment 3 in which we reversed the direction of the first sliding movement while we manipulated, across two conditions, whether the second movement contained continuous velocity information or not. Unlike the two previous experiments in which both movements approached the target locations and had therefore a similar connection to the distal goal, Experiment 3 creates an asymmetry between the two sequential movement components such that the movement directions are opposed and only the second movement is directed towards the distal goal. With this manipulation, we had two aims. First, we wanted to test whether participants rely exclusively on the velocity of a movement, regardless of its direction, when simulating the distal goal. Second, by reversing the first movement, we aimed at weakening its connection to the goal and, in contrast, to highlight the connection of the second one with respect to the goal. Based on this, we expected participants to be more likely to rely on movement-to-movement simulations in this experiment.

We predicted that if participants engage in movement-to-movement simulations, then they should be able to simulate the distal goal in the Sliding-Sliding condition, where they are presented with a second movement containing velocity information that they can feed into their simulation, but less so in the Sliding-Jumping condition, where this second movement containing velocity information is absent. However, if participants disregard the second movement and only engage in movement-to-goal simulations, then we do not expect a difference across the two conditions, as only observing the first movement should enable them to simulate the distal goal. Finally, since the reversed movements in the present study still preserved differences in peak velocity, we predicted that in both conditions participants would benefit from the exaggerations by producing more consistent motor-iconic mappings.

### Methods

**Participants.**   We recruited 50 participants (11 women; Age: $M$ = 31 years; $SD$ = 12.2 years) through Testable. The conditions of recruitment were identical to Experiment 1 and 2. We excluded one participant from our analyses. The design and analyses of this study were pre-registered on OSF (https://osf.io/3g24f).

**Stimuli.**   The full layout without the black occluder for both conditions is presented in Fig 8A. Participants saw a layout in which the box was initially displayed in the middle of the

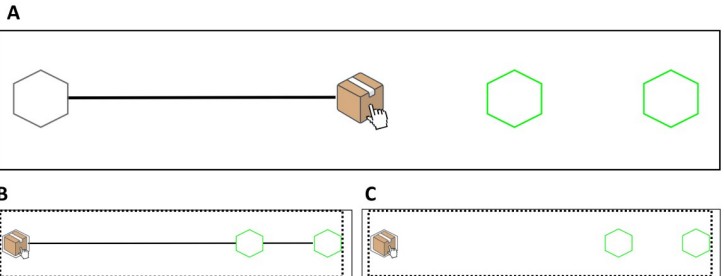

**Fig 8. Experimental layout used in Experiment 3. (A)** Participants in both conditions were first present with the full layout, without the occluder. Once the box was moved towards the intermediate location on the left, the black line changed differently depending on the condition. **(B)** In the Sliding-Sliding condition, the black line connected the intermediate location to the green target locations. **(C)** In the Sliding-Jumping condition, the black line disappeared from the display.

screen, at the rightmost end of the black line connecting the middle of the screen to the grey hexagonal location on the left-hand side of the screen.

During trials, once the box was moved from the starting location towards the grey location on the left, the black line along which the box had moved either got stretched out, thereby connecting the grey location to the green target locations on the right side of the screen (in the Sliding-Sliding condition, see Fig 8B) or completely disappeared from the display (in the Sliding-Jumping condition, see Fig 8C).

**Design and procedure.** Experiment 3 consisted of a mixed-factorial design with second movement type as a between-subject factor and degrees of exaggeration as a within-subject factor.

As in Experiments 1 and 2, participants were first presented with the full task layout, without the occluder (Fig 8A). In Experiment 3, however, the box moved continuously from the starting location now located in the middle of the screen, towards the intermediate target now located on the left. Participants saw two Normal sliding movements during familiarization. What happened next differed across the two between-subject conditions. In the Sliding- Sliding condition, once the box reached the intermediate target, the black line was stretched out and the box slid to either the near or far target location (Fig 8B). In the Sliding-Jumping condition the black line disappeared (Fig 8C). Then, the box disappeared and reappeared in either the near or far green target location. These changes in the line were introduced in order to highlight the differences in the second occluded movement. The disappearance or stretching of it before the onset of the second movement was used to inform participants about the upcoming disappearance-reappearance or sliding of the box towards the targets, respectively.

During trials, participants in both conditions saw the first sliding movement towards the grey target on the left, and saw either the disappearance of the line or its stretching before the occluder covered the entire layout. Note that the first movement participants saw was identical in both conditions; what differed during trials is the implied (i.e., occluded) second movement only.

## Results

**Consistency scores.** We conducted a 2x3 ANOVA on Consistency scores with second movement type as between subject factor and degrees of exaggeration as a within subject factor. The ANOVA yielded a significant main effect of degrees of exaggeration $F(2,94) = 35.9$, $p < .001$, $\eta^2 = .25$. The main effect of second movement type did not reach significance $F(1,47)$

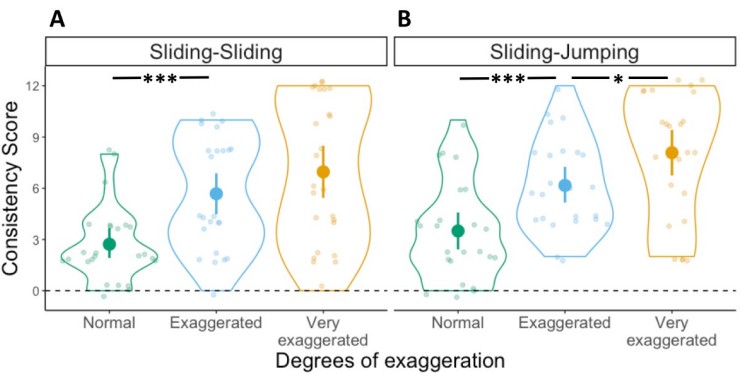

**Fig 9. Consistency scores in Experiment 3.** Distribution of Consistency scores in the (A) Sliding-Sliding and (B) Sliding-Jumping conditions. Violin plots represent the overall distribution of Consistency scores for each degree of exaggeration. The dashed horizontal line indicates a hypothetical value for random mapping (i.e., no consistency). Significant differences between degrees of exaggeration and across conditions as derived from a post-hoc t-test are indicated by horizontal lines and asterisks above the violin plots (* = $p \leq 0.05$; ** = $p \leq 0.01$; *** = $p \leq 0.001$).

= 1.34, $p$ = .25, $\eta^2$ = .016, nor did the interaction between these two factors F(2,94) = .18, $p$ = .81, $\eta^2$ = .002 (Fig 9).

**Mapping scores.** We conducted a 2x3 ANOVA with Mapping scores as dependent variable, second movement type as between-subject factor and degrees of exaggeration as within-subject factor. The ANOVA yielded main effects of second movement type ($F(1,47)$ = 8.3, $p$ = .006, $\eta^2$ = .11), degrees of exaggeration ($F(2,94)$ = 6.6, $p$ = .002, $\eta^2$ = .03) and an interaction between these factors ($F(2,94)$ = 3.5, $p$ = .03, $\eta^2$ = 02). Within the Sliding-Jumping condition, none of the pairwise comparisons across degrees of exaggeration using Bonferroni-corrected t-tests yielded a significant difference (all $t(94)$ > -1.1, $p$ >.86, $d$ < .09), whereas in the Sliding-Sliding condition both the comparison between Normal and Exaggerated ($t(94)$ = -3.4, $p$ = .003, $d$ = .87) and between Normal and Very Exaggerated ($t(94)$ = -4.1, $p$ < .001, $d$ = .88) differed significantly (Fig 10).

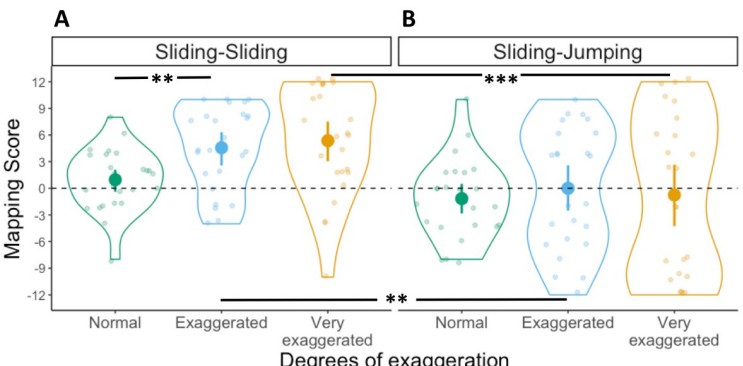

**Fig 10. Mapping scores in Experiment 3.** Distribution of Mapping scores in the (A) Sliding-Sliding and (B) Sliding-Jumping conditions, for the three degrees of exaggeration. Violin plots represent the overall distribution of Mapping scores for each degree of exaggeration. The dashed horizontal line indicates random mapping direction. Significant differences between degrees of exaggeration and across conditions as derived from a post-hoc t-test are indicated by horizontal lines and asterisks above the violin plots (* = $p \leq 0.05$; ** = $p \leq 0.01$; *** = $p \leq 0.001$).

## Discussion

In Experiment 3 we asked whether highlighting the connection of the second movement to the distal goal would affect how participants simulate a distal goal, and whether such change would also depend on whether the second movement contains velocity information or not. By doing this, we created a situation in which participants would be more likely to integrate the second movement into their simulations (i.e., movement-to-movement simulations). The findings of Experiment 3 suggest that participants do integrate information coming from the second movement component when simulating a distal goal. When presented with continuous movements containing velocity information which are then followed by another continuous movement containing velocity information (i.e., Sliding-Sliding), participants were more likely to map the movements to the target locations in a motor-iconic fashion. But when the same continuous movements were followed by a second discrete movement that did not contain velocity information (i.e., Sliding-Jumping), participants did not show a clear preference towards any mapping direction, and thus failed collectively to simulate the distal goal. Thus, our findings support a more complex conclusion regarding the simulations that observers use to predict a distal goal. In some (and maybe most) circumstances observers can simply rely on movements containing modulations in velocity to run direct movement-to-goal simulations that do not take into account a potential second movement. In others, however, the second movement, particularly when it contains informative kinematic features like velocity, can facilitate the process whereby observers link initial movements to their distal goals. In Experiment 3 we showed that one such situation concerns the reversal of the initial movement, which meant that the direction of the first and second movement were opposed, with the first going away and the second towards the target locations. This meant that, while the first movement still contained useful velocity information that could be used to simulate the distal goal, its change in direction led to a weakened relationship with the distal goal. Possibly, other situations in which the relationship between the movements or between the movements and the distal goal is modified might lead to a similar change in the type of simulations that observers rely on.

In sum, these results support the hypothesis that observers infer distal goals taking into account not only the kinematics of the first movement in a sequence, but also the features of an implied (i.e., occluded) second movement.

## General discussion

Previous research on SMC focused on settings in which observers made online predictions of unfolding movements in order to derive a proximal goal [42–44, 47]. Here we extend this focus to situations in which they need to simulate a future, more distal goal [53]. To do so, we first differentiated proximal and distal goals with respect to the number of motor acts used to achieve them: proximal goals are achieved by a single motor act, such as reaching for an object, while distal goals are achieved by means of at least two sequential motor acts, each with their own proximal (sub-)goal, like reaching for an object to then throw it [12, 13]. Following this distinction, we created a task in which we presented participants with the first component of a partially occluded two-step action sequence, and asked them to predict the distal goal of the sequence on the basis of communicative modulations present in the first component. Thus, our task differed from previous studies in SMC in that, instead of presenting one-step actions whose goals needed to be predicted as the movements unfold [see 8 for a review], we showed participants a full initial movement leading to an intermediate (sub-)goal, and asked them to simulate the distal goal of the entire sequence.

Using this paradigm, we were interested in studying the conditions under which observers can derive information about a distal goal by using two types of simulation: one in which observers use the kinematic information presented in an initial movement to simulate the distal goal directly, which we referred to as "movement-to-goal" simulation, and another in which they use that same kinematic information to simulate an upcoming movement, subsequently leading to a simulation of a distal goal and which we referred to as "movement-to-movement" simulation.

To do so, we first looked at whether the presence or absence of continuous velocity information in the first movement of the sequence would enable observers to simulate a distal goal. Participants in Experiment 1 were presented with either a continuous sliding movement that contained velocity information or a discrete jumping movement that only had information about its total duration. These movements were then followed by a second, occluded sliding movement towards one of two target locations. We found that participants established consistent mappings both when the velocity information was presented and when it was not, and that such mappings became more consistent with higher degrees of exaggeration, in line with previous findings in SMC [43, 53]. Despite the high consistency of their mappings, participants were only able to simulate a distal goal when the movements they observed contained modulations of velocity, and not when they only contained modulations in total duration, as indicated by the strong preference for motor-iconic mappings in the former but not the latter condition. This preference for motor-iconic mappings is a clear indication that observers are able to identify the underlying lawful relation that connects movements and distance during natural performance (i.e., aiming movements reach higher peak velocity when directed at further targets [15, 13], and use it to connect a first movement component in a sequence to its distal goal.

However, the findings of Experiment 1 were inconclusive with regard to the type of simulation used by participants to predict the distal goal. Participants may have integrated the second sliding movement into their interpretations, consistent with movement-to-movement simulations, or may have completely disregarded the implied second movement, thus supporting a more direct, movement-to-goal simulation. Thus, to get a better understanding of the type of simulation underlying participant's responses we conducted Experiment 2, where we presented participants with the same movements as in Experiment 1, but these were then followed by a movement that only contained information about duration. Thus, we created a situation in which participants would be less able to use movement-to-movement simulations to predict the distal goal of a movement containing velocity. The findings of Experiment 2 suggest that participants can engage in direct movement-to-goal simulations when observing movements containing velocity, and thus can bypass the second movement in their simulations of the distal goal. However, the results also showed that movement-to-goal simulations were leading to more variability in the way participants mapped the movements onto the target locations. This made us hypothesize that, at least in some circumstances, participants' simulations of a distal goal are sensitive to the presence of a second movement, even if this movement is not directly observed but only implied.

The particular aim of in Experiment 3 was to address this question. In other words, would there be situations in which movement-to-movement simulations also play a role? We hypothesized that one such circumstance would be if the relationship of the first movement with respect to the distal goal is weakened, thus highlighting the role played by the second movement. This weakening of the link between the movements was made possible by reversing the direction of the first movement, which was directed away from the target locations, while the second movement was directed towards them. With this manipulation, we expected participants to integrate the second movement into their simulations, and therefore also expected their responses to be affected by whether this second movement contained velocity

information or not. The results of Experiment 3 confirmed this prediction, as participants interpreted the movements presented differently depending on whether the second movement contained velocity information or not.

What does this set of studies tell us about the relationship between proximal and distal goals? One way of describing the relationship between proximal and distal goals, besides the number of motor acts leading to their achievement, is to locate them vertically along an action hierarchy, with overarching (potentially more distal) goals on top, and simpler motor acts (possibly more proximal) at the bottom [21, 59]. This way of describing proximal and distal goals is interesting for two reasons. First, it implies that proximal goals are sometimes instrumental for the achievement of more distal goals, as the former are simply the subgoals leading to the achievement of the latter (e.g., the proximal goal of picking up an apple is a subgoal leading to the more distal goal of eating it). Second, and maybe more relevant for our purposes, this hierarchical organization suggests that proximal goals located at different levels of the action hierarchy can be used by observers to simulate distal goals which might also vary along the same hierarchy. For example, the initial stages of an aiming movement towards a ball can then be used not only to predict whether the agent will then throw it into a large box [12] but more generally, whether the agent is doing so because she has a goal located high up in the action hierarchy, like tidying up a room or simply practicing her aim. In the context of SMC, where people exaggerate their movements to convey anticipatory information about their action goals and thus facilitate coordination, one would expect observers to be similarly able to derive information about such overarching, higher goals, especially when these are relevant to the achievement of a joint goal. For instance, observers are sensitive to the kinematics of their co-actor's instrumental movements while playing a speeded game, but even more so depending on whether the game is framed by the experimenter as a cooperative game, rather than a competitive one [14], suggesting that higher, and in this case pro-social, goals can have strong top-down effects on people's sensitivity to others' actions [see also 32, 41]. Within the context of SMC, observers have been shown to benefit from communicative modulations not only to predict simple action goals, but also to infer whether an actor is performing a given action sequence with the arguably more complex social goal of demonstrating the sequence to a naïve observer (i.e., teaching), or to coordinate with someone [45]. Altogether, these studies suggest that people can derive useful information from observing other people's movements, and from these observations infer more than mere action goals that vary with respect to the number of motor acts (i.e., proximal and distal), but that can also be located at different levels of an action hierarchy, starting from simple motor actions all the way up to complex social intentions.

An open question for future studies is whether communicative modulations of instrumental actions can, on top of facilitating the prediction and identification of more or less complex goals, trigger other types of inference about the observed action. For example, recent computational models of communicative demonstrations have shown that participants can learn the hidden reward structure of grid-like environments when observing movements that deviate from the most efficient trajectory (e.g., by visiting multiple tiles within a trial). Crucially, participants in this task increased their accuracy when told that the agent producing the movements knew that naïve observers would then watch and learn from them [60]. In line with findings from SMC and action observation, this indicates that observers interpret movements differently if they know that these were produced not only with an instrumental goal, but with a further communicative goal. Another interesting example of observers going beyond the mere prediction of goals is a recent study by Schmitz et al. [61], where they showed that observers can infer the hidden properties of an object (i.e., its weight) by relying on communicative modulations of reaching movements directed towards these objects. Thus, these two studies open up a venue for future research on people's capacity to derive other kinds of

information when observing communicative modulations, either about the person performing the action (e.g., what the person knows [62]) or about hidden properties of objects (e.g., their function [63]).

Extending our own previous research [53], we made two proposals about the underlying simulation processes that would enable observers to link communicative modulations in early instrumental movements to their distal goals. These two theoretical possibilities differ primarily in the amount of information needed in order for the simulation to occur. A "movement-to-goal" simulation is made possible when observers link an early movement to its distal goal directly, and thus do not need to take into account the role played by the second movement in the sequence. A "movement-to-movement" simulation, on the other hand, does integrate the second movement in the process of linking the early movement to the distal goal. The process of using the kinematic information present in an initial movement to feed into a simulation of a second movement is in some respects analogous to the process of action simulation described by Prinz & Rapinett [64], according to which observers use the early kinematic features present in a reaching movement before its occlusion to generate an internal simulation that replicates the kinematic features of the reaching movement and then applies them to the now extrapolated reaching movement occurring behind the occluder [65]. This internal simulation, which reuses information from the early stages of the movement, before its occlusion, enables observers to make accurate predictions about the reappearance of the movement after these short episodes of occlusion. As such, movement-to-movement simulations and the action simulation processes described by Prinz and Rappinet [64] seem to rely on the same underlying principle which enables observers to predict action trajectories and goals by means of internally regenerated movements, i.e. simulations. The main difference between the two processes, however, is that while action simulations are commonly used in the process of extrapolating partially occluded one-step movements, the movement-to-movement simulations we propose here are used to regenerate an entire movement, with its corresponding (distal) goal.

To address the question of how observers can simulate distal goals, we decided to focus on two-step action sequences in which only the first movement was visible to participants. This meant that, unlike previous research on SMC in which proximal movements are exaggerated to facilitate the online prediction of proximal goals, the first movement was temporally and spatially separated from the simulated distal goal. In our previous work [53], we discussed the possibility that such separation may have led some participants to change the way in which they interpreted the movements they saw. Specifically, participants in our experiments may have fully disregarded the instrumental aspects of the movement (i.e., delivering the box), while focusing exclusively on its communicative goal (i.e., informing observers about the upcoming delivery location). This could have happened despite the fact that participants were explicitly told that the movements had an instrumental goal at the beginning of the study. If participants fully disregarded the instrumental aspect of the movements, this would imply that they may have taken the movements as purely communicative movements that stand for, represent, or refer to particular target locations, similar to other "purely" communicative movements, like gestures, that are also said to stand for, represent, or refer to particular entities by means of hand and bodily movements. Indeed, functional accounts of gestures have recently put forward the idea that what makes a movement a "gesture" is the fact that these are bodily movements that are stripped away from their more habitual instrumental aspects (e.g., reaching and manipulating objects). In the process of becoming less "instrumental", these movements come to fulfill a different function, essential to most gestures: that of representing or referring to objects [66, 67], either for oneself or for others.

This way of understanding the difference between instrumental and communicative actions has interesting implications for our current findings and for SMC more generally. Given that SMC relies on the production and understanding of movements that have both an instrumental and communicative goal, this form of communication can be seen as occupying an intermediate position between fully instrumental and fully communicative movements [8–10, 60]. Consequently, one could argue that the communicative modulations present in SMC might already contain some of the ingredients that enable movements to become "representational", or "referential", thus making them more similar to gestures. For example, in our studies, communicative modulations of an early movement might be seen as referring to the distal goals of the sequence (corresponding, in this case, to specific movement endstates). What makes this relationship between movement and goal one of "reference" is the fact that the movements are separated, and thus "detached", from the goal. From the observer's point of view, this form of detachment might be seen as a first step in the process of interpreting movements as having the capacity to "represent", which subsequently might lead observers to come up with stable mappings between these movements and their goals. Whether and how SMC can provide a standpoint from which to study the relationship between instrumental and communicative actions, and how the former kind of action becomes more like the latter by means of gradually acquiring such "representational" features, are among some of the questions that will require further investigation.

Besides providing evidence that observers can simulate goals that are spatially and temporally removed from the here-and-now (i.e., that are distal), our studies also indicate that these simulations drive participants towards establishing specific mappings between movements and the distance of target locations, where faster movements are more likely to be mapped onto far locations [13]. In the Discussion of Experiment 1 we mentioned how this motor-iconic mapping can be contrasted with its reversal which, interestingly, was more likely to be present in the conditions in which participants saw movements that only contained information about their total duration (i.e., Jumping). In other words, participants in these conditions were more likely to interpret *longer durations* as directed to *further* target locations. As we pointed out, this mapping was also consistently used by participants in a previous study on SMC, by Vesper et al., [43], in which pairs of participants were asked to coordinate their actions by performing aiming movements towards one of three target locations. Their results show that Leaders systematically mapped shorter aiming movements to near targets and longer aiming movements to far targets.

Although our study and the one by Vesper and colleagues [43] differ in many respects, they both point to similarities in the intuitions that people have about the relationship between particular movement parameters (such as duration or velocity) and distances. Thus, in both studies, the predominantly chosen mappings can be considered instances of "motor-iconicity". Whereas most participants in the present study focused on the regularities between movement velocity and movement distance (i.e., faster velocities go with farther distances), the task layout in the Vesper et al. study highlighted the relationship between movement duration and movement distance (i.e., longer durations go with farther distances). What makes this latter mapping motor-iconic is the fact that the duration-distance mapping also originates from a regular relationship found in the performance of aiming movements, where people tend to take longer (in terms of duration) to reach further locations [68]. From this point of view, participants in the Jumping-Sliding and Jumping-Jumping conditions who, according to our interpretation, were "reversing" the mapping, might simply rely on a different, but still motor-iconic, relationship connecting the movements to their goals. Whether people are aware of these differences in motor-iconic relations, and whether they can use these to inform others about their goals, are among the questions that should be further investigated.

## Acknowledgments

We thank Fanni Takátsy for her help with data collection.

## Author Contributions

**Conceptualization:** Martin Dockendorff, Laura Schmitz, Cordula Vesper, Günther Knoblich.

**Data curation:** Martin Dockendorff.

**Formal analysis:** Martin Dockendorff.

**Funding acquisition:** Cordula Vesper, Günther Knoblich.

**Investigation:** Martin Dockendorff, Günther Knoblich.

**Methodology:** Martin Dockendorff, Laura Schmitz.

**Project administration:** Martin Dockendorff, Cordula Vesper, Günther Knoblich.

**Resources:** Günther Knoblich.

**Software:** Martin Dockendorff.

**Supervision:** Laura Schmitz, Cordula Vesper, Günther Knoblich.

**Validation:** Martin Dockendorff, Laura Schmitz, Cordula Vesper, Günther Knoblich.

**Visualization:** Martin Dockendorff.

**Writing – original draft:** Martin Dockendorff.

**Writing – review & editing:** Martin Dockendorff, Laura Schmitz, Cordula Vesper, Günther Knoblich.

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
