## [Decision Letter · Decision Letter 0]

6 Mar 2024

PONE-D-23-26818Communicative modulations of early action components support the prediction of distal goalsPLOS ONE

Dear Dr. Dockendorff,

Thank you for submitting your manuscript to PLOS ONE. After careful consideration, we feel that it has merit but does not fully meet PLOS ONE’s publication criteria as it currently stands. Therefore, we invite you to submit a revised version of the manuscript that addresses the points raised during the review process.

We look forward to receiving your revised manuscript.

Kind regards,

Alessia Tessari, Ph.D.

Academic Editor

PLOS ONE

Journal Requirements:

“This research was supported by the European Research Council under the European Union's Seventh Framework Program (FP7/2007-2013) / ERC grant agreement n° [609819], SOMICS.”

Additional Editor Comments (if provided):

Please, pay close attention to the comments and suggestions of the reviewers while preparing the new draft, and to the points raised by reviewer 1 particularly.

Reviewers' comments:

Reviewer's Responses to Questions

**Comments to the Author**

1. Is the manuscript technically sound, and do the data support the conclusions?

Reviewer #1: Partly

Reviewer #2: Yes

2. Has the statistical analysis been performed appropriately and rigorously? 

Reviewer #1: Yes

Reviewer #2: Yes

3. Have the authors made all data underlying the findings in their manuscript fully available?

Reviewer #1: Yes

Reviewer #2: Yes

4. Is the manuscript presented in an intelligible fashion and written in standard English?

Reviewer #1: Yes

Reviewer #2: Yes

5. Review Comments to the Author

Reviewer #1: The manuscript presents a set of three experiments in which the authors test the circumstances under which participants use kinematic information to infer a distal goal. Overall, the design was clear and the methods seem sound. The topic itself is quite interesting, and I think it makes an important contribution to the field. I mainly have some issues with the framing, and in particular the focus on “motor simulation” as the only potential explanation for the findings.

In the introductory text on Page 3, the authors refer to “coordination smoothers” as behavioral strategies, such as kinematic modulation, that can make an action more readable and thus facilitate prediction by an observer. I think it’s important, for the sake of a balanced discussion of this topic, to also note that kinematic modulation is likely not always a ‘strategy’ per se. In the example of piano duets, I would agree that this is likely driven by an intention to communicate/coordinate. However, when the authors go on to focus on the two-step action sequences such as grasping food to bring it to one’s mouth, I think it’s less clear that this would be an active strategy to signal what is being done. Particularly given that much of this work has used recordings of these reach-to-grasp movements that are used outside of any communicative or interactive context. Instead, these modulations are more likely to be a consequence of an action being embedded in a larger action-chain, and the motor system attuning to the context of the larger action. Observers can use the kinematics, but more as a cue, rather than an actively created signal. See, for example, the work by Runeson & Frykholm (1983). I see that further into the introduction the authors explicitly discuss action chains as an influence and communicative modulation as another, but then the two seem to be conflated again in the explanations of the experiments. For example, the authors write the following about Experiment 1: “If the presence of velocity information increases the likelihood that information from the first movement is interpreted as communicative, and thereby facilitates the prediction of a distal goal, participants should consistently map fast initial movements onto far targets and slow initial movements onto near targets”. I think it’s important to note that velocity information can be informative, without it being, or being perceived as, communicatively intended.

Page 6. In the discussion of motor representations, I believe the authors are appealing to a similar idea as what I outlined in my first comment, regarding the embedding of actions in the larger action chain. However, I think it can be made clearer why the distal goal would affect an earlier movement. To use the authors’ example, if there is a motor representation for throwing an object into a basket, presumably the representation involves picking up the object (1), and throwing it in such a way that its trajectory brings it into the basket (2). The representation of (2) will depend on the outcome of (1), such as the end position of the hand, but the text is currently unclear on why the kinematics of (1) would already be shaped by the representation of (2).

Page 10. The authors explain that the link between velocity profile and target distance as “motor iconic”, and likely resulting from motor representations. I wonder, why are motor representations the most likely explanation here? I guess this would involve an internal representation of moving an object, and matching that representation to the observation. Wouldn’t it be a much simpler explanation to say that statistical learning has taught us that high peak velocity is associated with longer/farther movements? The lawful relationship aspect is still there, but doesn’t require on-line motor mirroring. I think this is particularly important to make clear, because on page 12 there is the suggestion that there is either 1) movement-to-goal simulation, 2)movement-to-movement simulation, or 3) an arbitrary, spontaneously chosen link between observed kinematics and predicted goal that can be (as the authors say) more or less random. If there is a lawful relationship between information that can be perceived (i.e., velocity of an object) and a distal outcome, why do we need to simulate or represent a motor plan?

Page 20. The authors find differences between Exaggerated and Very Exaggerated movement profiles in the Mapping Score analyses. However, it was quite unclear to me on first reading which contrasts are being discussed. In other words, that there is a difference in mapping score of exaggerated and very-exaggerated movements when comparing sliding-sliding and jumping-sliding conditions (if I am understanding it correctly).

Reviewer #2: Dockendorff and co-authors, through three experiments, provide an interesting and original overview of how communicative modulations affect the prediction of distal goals. Their experiments nicely demonstrate that for observers to simulate distal goals via motor-iconic mappings, the initial movement needs to contain velocity information. Additionally, they examine whether participants engage in movement-to-movement or movement-to-goal simulations.

I believe the paper would benefit from the following adjustments:

- The main issue I see in the paper lies within Experiment 2. Here, the authors aim to investigate whether participants engage in movement-to-goal or movement-to-movement simulations. To address this question effectively, the focus should primarily be on comparing the sliding-sliding condition of Experiment 1 to the sliding-jumping condition in Experiment 2. While the authors do report these analyses, they seem to place more emphasis on the contrast between the sliding-jumping and jumping-jumping conditions. To strengthen the paper, I would suggest clarifying the relevance of the between-experiment comparison more strongly throughout the study. Additionally, providing a plot of this comparison would enhance the relevance of the findings.

- Throughout the experiment, the authors mention interindividual differences in the mapping style. For instance, on page 22, they claim that the lower motor-iconic mappings in the jumping-sliding condition (compared to the sliding-sliding condition) are likely due to some participants reversing the mappings (high consistency but reversed motor-iconic mappings). Similarly, on page 29, they mention that some participants reverse the mappings. I believe it is necessary to address this further. To strengthen these claims the authors could consider to:

1) Clarify why some participants show this behaviour. Is there any additional information about the sample analysed that might help explaining this pattern (e.g., questionnaires etc.)? Do the authors have any hypotheses?

2) Include result sections where discussing some of these interindividual differences. For instance, what is the percentage of people reversing the mappings?

3) Analyse the dispersion of the data. For instance, in line 636, the authors claim that there are fewer interindividual differences in the mapping style in the sliding-sliding condition compared to the sliding-jumping. Statistical analyses should be provided to support this claim.

4) Revise Fig4A. In line 636, they claim that interindividual differences seem to be entirely absent from the sliding-sliding condition of Experiment 1. However, upon examining the plot, there is at least one subject reversing the mapping. This should be clarified.

- Also, the authors should clarify why the timing at which the box remained stationary changes between experiments (Experiment 1 had a timing of 500ms [line 342] while Experiment 2 had a timing of 1500 ms [line 536]).

- In the introduction of Experiment 3, it would be beneficial to explain further why the authors chose to reverse the direction of the first movement, among other possible manipulations. Also, it would be beneficial to link it to existing evidence using the same manipulation.

- In the introduction, the authors largely discuss motor representations. If deemed relevant, there is interesting neuroimaging evidence linking motor representations and goals (i.e., rewards). It could be interesting to cite some brain-related studies in the introduction:

1) Adkins, T. J., & Lee, T. G. (2021). Reward modulates cortical representations of action. NeuroImage, 228, 117708. https://doi.org/10.1016/j.neuroimage.2020.117708

2) Galaro, J. K., Celnik, P., & Chib, V. S. (2019). Motor Cortex Excitability Reflects the Subjective Value of Reward and Mediates Its Effects on Incentive-Motivated Performance. The Journal of Neuroscience, 39(7), 1236–1248. https://doi.org/10.1523/JNEUROSCI.1254-18.2018

- Consider adding a practical example on page 12 to clarify what is meant by consistent "arbitrary" relationships between movements and distal goals.

- In line 562, correct the grammar (suggests, instead of suggest).

- Highlight significance bars in the plots.

Overall, Dockendorff et al.'s paper provides interesting insights into communicative modulations of actions. I believe that by addressing these points the paper will be clearer and stronger.

6. PLOS authors have the option to publish the peer review history of their article (what does this mean?). If published, this will include your full peer review and any attached files.

Reviewer #1: **Yes: **James Trujillo

Reviewer #2: **Yes: **Margherita Tecilla

---

## [Author Response · Author response to Decision Letter 0]

24 Apr 2024

Response to Reviewers:

Reviewer #1: The manuscript presents a set of three experiments in which the authors test the circumstances under which participants use kinematic information to infer a distal goal. Overall, the design was clear and the methods seem sound. The topic itself is quite interesting, and I think it makes an important contribution to the field. I mainly have some issues with the framing, and in particular the focus on “motor simulation” as the only potential explanation for the findings.

In the introductory text on Page 3, the authors refer to “coordination smoothers” as behavioral strategies, such as kinematic modulation, that can make an action more readable and thus facilitate prediction by an observer. I think it’s important, for the sake of a balanced discussion of this topic, to also note that kinematic modulation is likely not always a ‘strategy’ per se. In the example of piano duets, I would agree that this is likely driven by an intention to communicate/coordinate. However, when the authors go on to focus on the two-step action sequences such as grasping food to bring it to one’s mouth, I think it’s less clear that this would be an active strategy to signal what is being done. Particularly given that much of this work has used recordings of these reach-to-grasp movements that are used outside of any communicative or interactive context. Instead, these modulations are more likely to be a consequence of an action being embedded in a larger action-chain, and the motor system attuning to the context of the larger action. Observers can use the kinematics, but more as a cue, rather than an actively created signal. See, for example, the work by Runeson & Frykholm (1983). I see that further into the introduction the authors explicitly discuss action chains as an influence and communicative modulation as another, but then the two seem to be conflated again in the explanations of the experiments. For example, the authors write the following about Experiment 1: “If the presence of velocity information increases the likelihood that information from the first movement is interpreted as communicative, and thereby facilitates the prediction of a distal goal, participants should consistently map fast initial movements onto far targets and slow initial movements onto near targets”. I think it’s important to note that velocity information can be informative, without it being, or being perceived as, communicatively intended.

We agree with the distinction made by R1 between intentional signaling strategies and mere “cues”. Unlike signals, cues are not produced with the intention to communicate, but they can still be used by observers to understand (and predict) what an actor is doing. We now make this distinction more explicit throughout the Introduction, as well as in the transition from the first section, on action production/understanding, to the second section, on sensorimotor communication (lines 129-136). We finally highlight a key aspect of sensorimotor communication, kinematic exaggerations, which in some circumstances (e.g., joint actions) can be interpreted as intentional, in the sense of being produced with the goal of facilitating the observer’s predictions (lines 152-157)

Page 6. In the discussion of motor representations, I believe the authors are appealing to a similar idea as what I outlined in my first comment, regarding the embedding of actions in the larger action chain. However, I think it can be made clearer why the distal goal would affect an earlier movement. To use the authors’ example, if there is a motor representation for throwing an object into a basket, presumably the representation involves picking up the object (1), and throwing it in such a way that its trajectory brings it into the basket (2). The representation of (2) will depend on the outcome of (1), such as the end position of the hand, but the text is currently unclear on why the kinematics of (1) would already be shaped by the representation of (2).

We now clarify and simplify our use of the notion of “motor representation” in the Introduction (lines 108-116). In the case of action production, motor representations play a role in guiding actions towards their outcomes. Because of this guiding role, they can lead to visible changes in the kinematics of early movements, particularly when these movements are embedded in a larger action chain.

Page 10. The authors explain that the link between velocity profile and target distance as “motor iconic”, and likely resulting from motor representations. I wonder, why are motor representations the most likely explanation here? I guess this would involve an internal representation of moving an object, and matching that representation to the observation. Wouldn’t it be a much simpler explanation to say that statistical learning has taught us that high peak velocity is associated with longer/farther movements? The lawful relationship aspect is still there, but doesn’t require on-line motor mirroring. I think this is particularly important to make clear, because on page 12 there is the suggestion that there is either 1) movement-to-goal simulation, 2) movement-to-movement simulation, or 3) an arbitrary, spontaneously chosen link between observed kinematics and predicted goal that can be (as the authors say) more or less random. If there is a lawful relationship between information that can be perceived (i.e., velocity of an object) and a distal outcome, why do we need to simulate or represent a motor plan?

We agree with R1 that, although our results are consistent with on-line motor mirroring, they do not provide any direct evidence for it. We also agree that other processes (e.g., statistical learning, action-effect associations) can also account for how a regular relationship between movements and target distance is established. We now make this point explicit in the Introduction, where we also clarify our use of the term “simulation” (lines 232-237). We use “simulation” in a broad sense to refer to the capacity observers have to link the kinematics of early movements to distal goals.

Page 20. The authors find differences between Exaggerated and Very Exaggerated movement profiles in the Mapping Score analyses. However, it was quite unclear to me on first reading which contrasts are being discussed. In other words, that there is a difference in mapping score of exaggerated and very-exaggerated movements when comparing sliding-sliding and jumping-sliding conditions (if I am understanding it correctly).

Yes, the comparisons are across conditions, suggesting that only in the Sliding-Sliding condition participants were able to map consistently in a motor-iconic fashion. We now make this comparison more explicit in the Results section (lines 422-423). We also add significance bars in each Figure.

Reviewer #2: Dockendorff and co-authors, through three experiments, provide an interesting and original overview of how communicative modulations affect the prediction of distal goals. Their experiments nicely demonstrate that for observers to simulate distal goals via motor-iconic mappings, the initial movement needs to contain velocity information. Additionally, they examine whether participants engage in movement-to-movement or movement-to-goal simulations.

I believe the paper would benefit from the following adjustments:

- The main issue I see in the paper lies within Experiment 2. Here, the authors aim to investigate whether participants engage in movement-to-goal or movement-to-movement simulations. To address this question effectively, the focus should primarily be on comparing the sliding-sliding condition of Experiment 1 to the sliding-jumping condition in Experiment 2. While the authors do report these analyses, they seem to place more emphasis on the contrast between the sliding-jumping and jumping-jumping conditions. To strengthen the paper, I would suggest clarifying the relevance of the between-experiment comparison more strongly throughout the study. Additionally, providing a plot of this comparison would enhance the relevance of the findings.

We now clarify the between-experiment comparison in the Results section of Experiment 2, specifically in the section titled “Type of simulation” (lines 610-625). There, we explain how the comparison between Sliding-Sliding (of Exp 1) and Sliding-Jumping (of Exp 2) can be used to inform us about the type of simulation that participants are using to connect the movements to their goals. We also expand on this point in the Discussion of Experiment 2 (lines 652-665).

- Throughout the experiment, the authors mention interindividual differences in the mapping style. For instance, on page 22, they claim that the lower motor-iconic mappings in the jumping-sliding condition (compared to the sliding-sliding condition) are likely due to some participants reversing the mappings (high consistency but reversed motor-iconic mappings). Similarly, on page 29, they mention that some participants reverse the mappings. I believe it is necessary to address this further. To strengthen these claims the authors could consider to:

1) Clarify why some participants show this behaviour. Is there any additional information about the sample analysed that might help explaining this pattern (e.g., questionnaires etc.)? Do the authors have any hypotheses?

As noted by R2, we proposed that participants who reversed the mapping were choosing it at the beginning of the experiment. We also proposed, briefly in the General Discussion, an alternative explanation for this reversal, according to which some participants had a different intuition about the relationship that connects movement duration and target distance. We now highlight this alternative explanation already in the Discussion of Experiment 1 (lines 474-486) and expand it further in the General Discussion (lines 1012-1037). Indeed, as revealed by participant’s self-reports, observing movements that had only differences in total duration (i.e., Jumping), led some of them to rely on the intuition that longer movements (i.e., movements with longer total durations) are more likely to travel longer distances (i.e., further target locations). Since longer movements have, in our task, a lower velocity, this intuition leads to a reversal of the mapping.

2) Include result sections where discussing some of these interindividual differences. For instance, what is the percentage of people reversing the mappings?

We now add the total number of participants reversing the mapping in the Discussion of Experiments 1 (for Jumping-Sliding see lines 471-472) and 2 (for Sliding-Jumping, see lines 662-665) We also discuss in detail the reasons why we think this split may have occurred (see our previous comment and lines 474-486 and 1012-1037)

3) Analyse the dispersion of the data. For instance, in line 636, the authors claim that there are fewer interindividual differences in the mapping style in the sliding-sliding condition compared to the sliding-jumping. Statistical analyses should be provided to support this claim.

As indicated in our previous answer, we now provide more detailed information about the number of participants who reversed the mapping as well as the reason for why some participants did this. We removed the claim about inter-individual differences, as this was not the reason why we decided to conduct Experiment 3. Instead, we now focus on simply describing the pattern of results in the Sliding-Jumping condition (lines 659-665). We hope that these changes make our claims more clear.

4) Revise Fig4A. In line 636, they claim that interindividual differences seem to be entirely absent from the sliding-sliding condition of Experiment 1. However, upon examining the plot, there is at least one subject reversing the mapping. This should be clarified.

We corrected the wording in the text. R2 is right in pointing out that one participant reversed the mapping in Sliding-Sliding. Now, instead of arguing that these interindividual differences seem to be entirely absent, we now simply mention the number of participants reversing the mapping in Experiment 1 (lines 471-472) and in the Sliding-Jumping of Experiment 2 (lines 662-665)

- Also, the authors should clarify why the timing at which the box remained stationary changes between experiments (Experiment 1 had a timing of 500ms [line 342] while Experiment 2 had a timing of 1500 ms [line 536]).

This was a mistake. The time the box remained stationary was 1500 ms in both experiments. We changed this now (line 336)

- In the introduction of Experiment 3, it would be beneficial to explain further why the authors chose to reverse the direction of the first movement, among other possible manipulations. Also, it would be beneficial to link it to existing evidence using the same manipulation.

We reversed the first movement to see if one of the simulations we propose (i.e., movement-to-movement simulation) can still play a role in how participants predict distal goals. As we now explain at the beginning of Experiment 3 (lines 684-690) the reversal of the first movement had two aims: weakening the relationship between the first movement and the distal goal (and thus, strengthening the one between the second movement and the distal goals), and furthermore, by maintaining the velocity information like in Experiment 1 and 2, verify that participants rely on velocity, regardless of the direction of the movement.

- In the introduction, the authors largely discuss motor representations. If deemed relevant, there is interesting neuroimaging evidence linking motor representations and goals (i.e., rewards). It could be interesting to cite some brain-related studies in the introduction:

1) Adkins, T. J., & Lee, T. G. (2021). Reward modulates cortical representations of action. NeuroImage, 228, 117708. https://doi.org/10.1016/j.neuroimage.2020.117708

2) Galaro, J. K., Celnik, P., & Chib, V. S. (2019). Motor Cortex Excitability Reflects the Subjective Value of Reward and Mediates Its Effects on Incentive-Motivated Performance. The Journal of Neuroscience, 39(7), 1236–1248. https://doi.org/10.1523/JNEUROSCI.1254-18.2018.

We thank R2 for these suggestions. They are indeed relevant. We have added them to the Introduction, specifically when we discuss the effect of distal goals on early kinematics (line 114-116). Specifically, we describe the expected value (i.e., rewards) of goals as another feature of distal goals.

- Consider adding a practical example on page 12 to clarify what is meant by consistent "arbitrary" relationships between movements and distal goals.

We now clarify in what sense we use “arbitrary” (lines 242-247). By arbitrary, we simply mean that participants choose, at the beginning of the experiment, a mapping, and they stick to it throughout the study. In other words, they understand that they should establish consistent links between movement and distance, but they do not rely on regular relationships (i.e., motor-iconicity) to do so.

- In line 562, correct the grammar (suggests, instead of suggest).

We corrected this now (line 578)

- Highlight significance bars in the plots.

We added significance bars and asterisks to indicate when the comparison between distributions is significant, as well as the strength of the significance test (see Figures and Figure captions)

Overall, Dockendorff et al.'s paper provides interesting insights into communicative modulations of actions. I believe that by addressing these points the paper will be clearer and stronger.

---

## [Decision Letter · Decision Letter 1]

29 May 2024

PONE-D-23-26818R1Communicative modulations of early action components support the prediction of distal goalsPLOS ONE

Dear Dr. Dockendorff,

Thank you for submitting your manuscript to PLOS ONE. After careful consideration, we feel that it has merit but does not fully meet PLOS ONE’s publication criteria as it currently stands. Therefore, we invite you to submit a revised version of the manuscript that addresses the points raised during the review process.

We look forward to receiving your revised manuscript.

Kind regards,

Alessia Tessari, Ph.D.

Academic Editor

PLOS ONE

Journal Requirements:

Additional Editor Comments:

Please, pay particular attention to Reviewer 1's requests.

Reviewers' comments:

Reviewer's Responses to Questions

**Comments to the Author**

1. If the authors have adequately addressed your comments raised in a previous round of review and you feel that this manuscript is now acceptable for publication, you may indicate that here to bypass the “Comments to the Author” section, enter your conflict of interest statement in the “Confidential to Editor” section, and submit your "Accept" recommendation.

Reviewer #1: (No Response)

Reviewer #2: (No Response)

2. Is the manuscript technically sound, and do the data support the conclusions?

Reviewer #1: Yes

Reviewer #2: Yes

3. Has the statistical analysis been performed appropriately and rigorously? 

Reviewer #1: Yes

Reviewer #2: Yes

4. Have the authors made all data underlying the findings in their manuscript fully available?

Reviewer #1: Yes

Reviewer #2: Yes

5. Is the manuscript presented in an intelligible fashion and written in standard English?

Reviewer #1: Yes

Reviewer #2: Yes

6. Review Comments to the Author

Reviewer #1: Overall, the authors did a good job of addressing the comments that I had in the first round. The manuscript reads much better now. I have a couple of points that came up during this reading that I think would make the manuscript much stronger in how the results are interpreted.

Lines 622-625: the authors interpret the non-significant second-movement type effect as demonstrating that participants do not need the velocity information of the second movement, and thus that “participants can rely on direct movement-to-goal simulation”. However, a non-significant p-value should not be taken as evidence for a lack of effect. Given that this outcome is important for the overall interpretation of the experimental effects, the authors should conduct an additional analysis to determine if there is indeed evidence for a lack of effect, for example using a Bayesian approach.

Experiment 3. On re-reading the discussion of these results, I wonder about the conclusion that the authors draw: “while the first movement still contained useful velocity information that could be used to simulate the distal goal, its change in direction led to a weakened relationship with the distal goal”. The authors acknowledge that the change in direction weakened the link between the first movement and the goal, but I wonder if it did so to the point that the connection becomes arbitrary. In a Gestalt framing of these experiments, the two movements are seen as one sequence or action chain due to their similarity in direction. But with this similarity in direction being removed, these two movements would not likely show the type of Gestalt coherence that would link them as being part of one goal-directed sequence. The seemingly arbitrary mapping between goal and first movement could be learned through perceptual experience, but not if there is no feedback. I don’t suggest that the authors add any additional experiments or anything to that extent, but I think that this (or a similar) interpretation should also be presented as a possible alternative.

Minor:

Lines 803-807: this is quite a long sentence with a lot of information. I think it would be helpful to break this up a bit.

Reviewer #2: I thank Dockendorff et al. for thoroughly addressing the points I raised in my comments.

I still find a minor incongruence in Experiment 2. In lines 577-578, the authors reported that “However, there was no significant main effect of the first movement type.” Yet, in lines 646-648, they wrote, “This was indicated by the higher consistency and higher number of motor-iconic mappings in the Sliding-Jumping condition compared to the Jumping-Jumping condition.” Could the authors clarify this? Perhaps the authors should limit their conclusions to the mapping scores.

I appreciated that the authors included significance bars in the plots; however, this is limited to within-subject factors. I believe it would be relevant to include bars to highlight significant results on the between-subject factors as well, to communicate the findings better.

7. PLOS authors have the option to publish the peer review history of their article (what does this mean?). If published, this will include your full peer review and any attached files.

Reviewer #1: **Yes: **James Trujillo

Reviewer #2: **Yes: **Margherita Tecilla

---

## [Author Response · Author response to Decision Letter 1]

7 Jun 2024

Reviewer #1: Overall, the authors did a good job of addressing the comments that I had in the first round. The manuscript reads much better now. I have a couple of points that came up during this reading that I think would make the manuscript much stronger in how the results are interpreted.

Lines 622-625: the authors interpret the non-significant second-movement type effect as demonstrating that participants do not need the velocity information of the second movement, and thus that “participants can rely on direct movement-to-goal simulation”. However, a non-significant p-value should not be taken as evidence for a lack of effect. Given that this outcome is important for the overall interpretation of the experimental effects, the authors should conduct an additional analysis to determine if there is indeed evidence for a lack of effect, for example using a Bayesian approach.

A:We clarify this in the Discussion section of Experiment 1 (lines 677-686), where we now make more explicit our interpretation of our findings. More specifically, we conclude that the findings of Experiment 2, together with those of Experiment 1, are consistent with the possibility that both types of simulation (movement-to-movement and movement-to-goal) can play a role in how participants link early communicative modulations to distal goals.

Experiment 3. On re-reading the discussion of these results, I wonder about the conclusion that the authors draw: “while the first movement still contained useful velocity information that could be used to simulate the distal goal, its change in direction led to a weakened relationship with the distal goal”. The authors acknowledge that the change in direction weakened the link between the first movement and the goal, but I wonder if it did so to the point that the connection becomes arbitrary. In a Gestalt framing of these experiments, the two movements are seen as one sequence or action chain due to their similarity in direction. But with this similarity in direction being removed, these two movements would not likely show the type of Gestalt coherence that would link them as being part of one goal-directed sequence. The seemingly arbitrary mapping between goal and first movement could be learned through perceptual experience, but not if there is no feedback. I don’t suggest that the authors add any additional experiments or anything to that extent, but I think that this (or a similar) interpretation should also be presented as a possible alternative.

A:The purpose of our manipulation of direction in Exp 3 was indeed to weaken the relationship of the first movement with the second movement. However, our data suggest that this did not make the connection arbitrary: The majority of participants still chose a motor iconic mapping in the Sliding-Sliding condition (see Figure 10A)

Regarding the suggestion that a pure learning account could also explain our results: It is important to remember that, during trials, participants never saw the box actually being delivered to the near or far target. Learning by association, we think, could only be between first and second movement, but not the distal goal. As such, we would like to point towards our existing discussion of alternatives to a simulation account (lines 235-237), where we mention statistical learning as a theoretical possibility. We also now include Gestalt-like grouping principles (e.g., similarity) as a theoretical possibility.

Minor:

Lines 803-807: this is quite a long sentence with a lot of information. I think it would be helpful to break this up a bit.

A:We did this now (see lines 822-826).

Reviewer #2: I thank Dockendorff et al. for thoroughly addressing the points I raised in my comments.

I still find a minor incongruence in Experiment 2. In lines 577-578, the authors reported that “However, there was no significant main effect of the first movement type.” Yet, in lines 646-648, they wrote, “This was indicated by the higher consistency and higher number of motor-iconic mappings in the Sliding-Jumping condition compared to the Jumping-Jumping condition.” Could the authors clarify this? Perhaps the authors should limit their conclusions to the mapping scores.

A:We thank R2 for pointing out this incongruence. Indeed, there was no significant main effect of first movement type. We corrected this in the Discussion of Experiment 2 (lines 653-655), where we now limit our interpretation to motor-iconic mappings. We also point out that in the Sliding-Jumping condition consistency increased as a function of degrees of exaggeration.

I appreciated that the authors included significance bars in the plots; however, this is limited to within-subject factors. I believe it would be relevant to include bars to highlight significant results on the between-subject factors as well, to communicate the findings better.

A:We now added the significance bars across conditions whenever the differences between distributions are significant.

---

## [Editor Report · Decision Letter 2]

12 Jun 2024

Communicative modulations of early action components support the prediction of distal goals

PONE-D-23-26818R2

Dear Dr. Dockendorff,

We’re pleased to inform you that your manuscript has been judged scientifically suitable for publication and will be formally accepted for publication once it meets all outstanding technical requirements.

Kind regards,

Alessia Tessari, Ph.D.

Academic Editor

PLOS ONE
---

## [Editor Report · Acceptance letter]

18 Jun 2024

PONE-D-23-26818R2 

PLOS ONE

Dear Dr. Dockendorff, 

I'm pleased to inform you that your manuscript has been deemed suitable for publication in PLOS ONE. Congratulations! Your manuscript is now being handed over to our production team.

Kind regards, 

on behalf of

Professor Alessia Tessari 

Academic Editor

PLOS ONE